# Improved Aerobic Capacity and Adipokine Profile Together with Weight Loss Improve Glycemic Control without Changes in Skeletal Muscle GLUT-4 Gene Expression in Middle-Aged Subjects with Impaired Glucose Tolerance

**DOI:** 10.3390/ijerph19148327

**Published:** 2022-07-07

**Authors:** Mika Venojärvi, Jaana Lindström, Sirkka Aunola, Pirjo Nuutila, Mustafa Atalay

**Affiliations:** 1Institute of Biomedicine, Sport and Exercise Medicine, University of Eastern Finland, 70210 Kuopio, Finland; 2Population Health Unit, Department of Public Health and Welfare, Finnish Institute for Health and Welfare, 00271 Helsinki, Finland; jaana.lindstrom@thl.fi; 3Functional Capacity Unit, Department of Health, Functional Capacity and Welfare, National Institute for Health and Welfare, 20740 Turku, Finland; sirkka.aunola@gmail.com; 4PET Centre, University of Turku, 20500 Turku, Finland; pirjo.nuutila@utu.fi; 5Institute of Biomedicine, Physiology, University of Eastern Finland, 70210 Kuopio, Finland; mustafa.atalay@uef.fi

**Keywords:** aerobic capacity, glycemic control, impaired glucose tolerance, adipokines, physical activity

## Abstract

(1) Objective: The aim of this study was to clarify the role of adipokines in the regulation of glucose metabolism in middle-aged obese subjects with impaired glucose tolerance in response to a long-term exercise and dietary intervention. (2) Methods: Skeletal muscle, plasma and serum samples were examined in 22 subjects from an exercise–diet intervention study aiming to prevent type 2 diabetes. The subjects were further divided into two subgroups (non-responders n = 9 and responders n = 13) based on their achievement in losing at least 3 kg. (3) Results: The two-year exercise–diet intervention reduced leptin levels and increased adiponectin levels in responders; the changes in leptin levels were significantly associated with changes in their weights (r = 0.662, *p* < 0.01). In responders, insulin sensitivity (Bennett and McAuley index) increased and was associated with changes in maximal oxygen uptake (VO_2_peak) (r = 0.831, *p* < 0.010 and r = 0.890, *p* < 0.01). In addition, the VO_2_peak and oxidative capacity of skeletal muscle improved in responders, but not in non-responders. However, there were no changes between the two groups in expressions of the glucose transporter protein-4 (GLUT-4) gene or of AMP-activated protein kinase (AMPK)-α1 or AMPK-α2 proteins. (4) Conclusions: The exercise–diet intervention decreased serum leptin and increased serum adiponectin concentrations, improved glucose control without affecting GLUT-4 gene expression in the skeletal muscle in responders.

## 1. Introduction

Impaired insulin sensitivity in skeletal muscle is an early defect in the pathogenesis of type 2 diabetes since it can be detected several years before the development of overt diabetes [1]. Insulin resistance in skeletal muscle is associated with an elevated adipose tissue mass [2]. The interplay between adipose tissue and skeletal muscle is important as the body tries to defend itself against insulin resistance in the peripheral tissues; this interplay is regulated by several factors and through more than one pathway [3]. Earlier studies revealed that adipose tissue secretes more than 50 different adipokines into circulation [2], while recent proteomic studies identified even more adipokines being secreted into the circulation and exosomes [4]. It is well-known that several adipokines and cytokines, including adiponectin, leptin, interleukin-6 (IL-6), monocyte chemoattractant protein-1 (MCP-1), INF-γ (interferon-γ) and tumor necrosis factor-α (TNF-α), can influence insulin signaling in skeletal muscle [3]. The down-regulation of the insulin receptor substrate 1 (IRS-1) pathway by TNF-α and a reduced activation of 5′-AMP activated protein kinase and AMPK (AMP-activated protein kinase) due to leptin resistance have been associated with insulin resistance [5,6]. IL-6 induces the hepatic synthesis of C-reactive protein (CRP) and its levels have been linked with visceral adiposity [7] but its role in obesity and insulin resistance is controversial. Monocyte chemoattractant protein-1 (MCP-1) also aggravates insulin resistance at doses similar to its physiological plasma concentrations (200 pg/mL) [2]. In contrast to the other adipokines, adiponectin concentrations are low in obesity and at the stage of insulin resistance; therefore, adiponectin is considered to be an insulin-sensitizing hormone [8]. Skeletal muscle cytokine expression can be increased via several pathways, including the p38MAPK and NF-kB pathways; this pattern of inflammatory expression seems to depend on the duration of obesity and the stage of type 2 diabetes. It has been reported that the INF-γ pathway is up-regulated in skeletal muscle cells isolated from insulin-resistant type 2 diabetic patients [9]. On the other hand, rodent studies indicated that *INF-*γ^−/−^ mice displayed better glucose tolerance compared to their wild type counterparts [10].

There are both human and animal studies demonstrating that long-term physical activity increased the expression of the GLUT-4 gene in skeletal muscle in middle-aged active subjects as compared to their sedentary peers [11]. Furthermore, in rats fed a high-fat diet, the expression of the GLUT-4 gene was decreased in skeletal muscle [12]. Nevertheless, as far as we are aware, there is no information on the effects of a long-term exercise–diet intervention on GLUT-4 and IRS-1 gene expression.

Skeletal muscle is a major site of insulin-stimulated glucose disposal [13]. In addition to insulin, physical exercise (i.e., muscle contraction and hypoxia) stimulates the translocation of the glucose transporter protein-4 (GLUT-4) to the plasma membrane. These two stimuli, i.e., insulin and physical exercise, act at least partially via independent pathways. These alternative pathways, which act in an additive manner to insulin-stimulated glucose uptake, could potentially improve the glycemic control of insulin-resistant subjects. AMPK is one of the key players in the pathway stimulated by muscle contraction [14]. The increased ratio of AMP to ATP activates this energy sensor protein, which is heterotrimetric and consists of a catalytic α and regulatory β and γ subunits [15]. The AMPK isoforms α1 and α2 are expressed in skeletal muscle [16] and their activities can be increased by moderate-intensity exercise but not with low-intensity exercise equal to or below 40% of VO_2_max [17]. Thus, AMPK also plays a key role in the regulation of fatty acid oxidation in skeletal muscle and it can be activated by adiponectin and leptin, resulting in enhanced fatty acid oxidation and glucose uptake [14]. In addition, the systemic effects of increased AMPK activity are regulated by many factors, i.e., adiponectin, insulin, IL-6, leptin and TNF-α. Understanding how these secreted signaling proteins regulate AMPK activity may yield clues to discovering novel therapeutic strategies for metabolic disorders, such as diabetes, insulin resistance and obesity [15]. It has also been debated whether exercise can improve insulin sensitivity in overweight Individuals without accompanying weight loss [18,19]. Dube and co-workers reported that both diet-induced weight loss and exercise training improved insulin resistance in overweight-to-obese adults [18]. Church et al. [20] revealed that, in patients with type 2 diabetes, the combination of aerobic and resistance training improved glycemic control and maximum oxygen consumption when compared to the control group. The same results could not be achieved by either aerobic or resistance training alone [21]. In this study, we explored the effects of a 2-year exercise–diet intervention on glucose metabolism in middle-aged obese subjects with impaired glucose tolerance (IGT) and separately in those who achieved or did not achieve a weight loss during this two year period. Here, we also report the responses of plasma levels of adipokines and cytokines to our long-term exercise and dietary intervention to provide evidence of new approaches that could be beneficial in the prevention and treatment of type 2 diabetes.

## 2. Materials and Methods

This is a sub-study of the Finnish Diabetes Prevention Study (DPS), a randomized controlled trial aiming to prevent type 2 diabetes in individuals with IGT at baseline, details of which have been described elsewhere [22,23]. IGT was defined by OGTT (oral glucose tolerance test) [18]. The sub-study was carried out in one of the five study clinics (Turku). Blood samples for metabolic indices were taken at baseline and after 2 years. A maximal exercise test and muscle sampling (with no strenuous exercising during the preceding 2 days) were performed at 6 months and after 2 years. The Ethical Committee of the Hospital District of South-West Finland, Turku, Finland and the Ethics Committee of the Rehabilitation Research Centre of the Social Insurance Institution of Finland approved the protocol of the sub-study (25 January 1996). On 6 March 1992, the Ethics Committee of The National Public Health Institute in Helsinki, Finland approved the protocol of the main study, and all the study subjects gave their written informed consent.

### 2.1. Subjects

A total of 110 obese subjects with impaired glucose tolerance, based on two oral glucose tolerance tests, were randomized to the intervention or the control group in the Turku clinic. Of the intervention group participants, the 22 subjects who volunteered to provide samples from the *m. vastus lateralis* were included in this study. The subjects were retrospectively divided into two subgroups, non-responders (n = 9) and responders (n = 13) according to their weight loss at the year 2 assessment point. The subgroup of non-responders was composed of subjects in whom weight loss was equal to or less than 1.3 kg (weight change range +3.0–−1.3 kg; mean ± std +0.33 ± 0.6 kg) and the subgroup of the responders was composed of subjects achieving at least a weight loss of 2.9 kg (range −2.9–−14.5 kg; mean ± std −7.7 ± 1.0 kg) during the intervention (Figure 1). The characteristics of the subjects are shown in Table 1.

### 2.2. Intervention

The lifestyle intervention started with intensive dietary counselling [24]. After 6 months, a supervised, progressive and individually tailored circuit-type resistance training program started. The main goals of the intervention were as follows: (1) weight reduction of 5% or more, (2) less than 30% of the daily energy intake from fat, (3) less than 10% of the daily energy intake from saturated fat, (4) fiber intake of 15 g per 1000 kcal or more and (5) physical activity at moderate intensity for 30 min or more each day. The implementation of the intervention program was previously reported [23,24]. Briefly, the participants in the intervention group were given detailed and individualized counselling to achieve the lifestyle goals set for them. They had seven individual counselling sessions with a nutritionist during the first year and every three months thereafter. During the first 6 months, the intervention focused on dietary counselling, in order to allow the subjects to concentrate on changing their eating habits. The subjects were also individually encouraged by a nutritionist to increase their physical activity, and later on, to participate both in regular resistance training at a gym as well as undertaking aerobic exercise. After 6 months, a supervised exercise training session arranged twice a week was added to the intervention program, as previously described [25]. The supervised training was progressive and individually designed, consisting mostly of strength and power training in the gym, along with spinning exercises and aerobic gymnastic exercises. The power-type strength training was organized as circuit training. Any physical activity lasting 30 min or longer was recorded by the participants. The subjects were advised to exercise with moderate-to-intense effort for at least 30 min per session and three to four times a week. The content of training program was described in greater detail in our previous report [25].

### 2.3. Muscle Biopsy

A skeletal muscle biopsy was taken from the vastus lateralis muscle under local anesthesia (lidocaine 10 mg/mL), using the ‘semi-open’ conchotomy technique [26]. Muscle samples were immediately frozen in liquid nitrogen and stored at −80 °C until analysis. Samples for biochemical analyses were thawed in an ice bath, weighed, and homogenized in 1:50 (*w*/*v*) of 1 M Tris buffer pH adjusted to 7.5 in a manually operated all-glass homogenizer [27].

### 2.4. Total RNA Preparation

Total RNA was extracted from ~20 mg of muscle pieces using a mono-phasic solution of phenol and guanidine isothiocyanate (TRIzol^®^ Reagent; Total RNA Isolation Reagent, Life Technologies, GIBCO BRL, Rockville, MD, USA) and chloroform, isopropyl alcohol, 75% ethanol and RNase-free water. The isolated RNA was further purified with the GenElute Mammalian Total RNA kit (Sigma Chemicals, St Louis, MO, USA). The absorption ratio (260:280 nm) varied from 1.6 to 2.0.

### 2.5. Real Time PCR

The first-strand cDNA was synthesized from 0.1 μg of total RNA using SuperScript™ II RNase H-Reverse Transcriptase (Invitrogen, Waltham, MA, USA, Life Technologies, Carlsbad, CA, USA). Primers and probes (GLUT-4, INF-γ, IRS-1 and TNF-α) were designed using Primer Express software (PE Biosystems, Foster City, CA, USA) (Table 2). Real-time (RT) PCR was performed using the ABI PRISM 7700 Sequence Detector (Applied Biosystems, Foster City, CA, USA). The probes were labelled with FAM at the 5′ end as the reporter dye and TAMRA at the 3′ end as the quencher dye. Gene expression of the gene of interest (mRNA) was calculated using the ∆-critical threshold method, normalized to GAPDH (glyceraldehyde-3-phosphate dehydrogenase) and expressed as post- vs. pre-values in non-responders and responders [28].

### 2.6. Immunoblotting

One-dimensional sodium dodecyl sulphate polyacrylamide vertical gel electrophoresis was performed to separate proteins according to their molecular weights [25]. The percentage of polyacrylamide was 10%. After separation, proteins were transferred to nitrocellulose membranes (0.45 mm thick, PROTRAN^®^, San Francisco, CA, USA) in a Bio-Rad trans-blot electrophoretic transfer cell at a constant voltage of 100 V for 1 h. After protein transfer, the nitrocellulose membranes were blocked for 1 h with 5% non-fat dry milk at +37 °C. The blots were incubated overnight at +4 °C with the following antibodies: goat polyclonal anti-AMPK-α1 and -AMPK-α2 (Santa Cruz, CA, USA). Horseradish peroxidase-conjugated immunoglobulins were used as secondary antibodies (Santa Cruz, CA, USA). The membranes were developed with an enhanced chemiluminescence method (NEN Life Sciences, Boston, MA, USA) and quantified using image analysis software (NIH-Image, Frederick, MD, USA). The results were normalized with beta-actin values (LabVision/NeoMarkers, Fremont, CA, USA: MS-742-S0). The protein concentrations were measured from homogenates using the BCA method (Pierce, Rockford, IL, USA) after equal amounts of the protein extract (20 μg) were loaded onto the gels.

### 2.7. Assessment of Enzyme Activities in the Muscle Samples

The citrate synthase (CS, E.C.4.1.3.7) activity was assayed from 50 μL of the homogenate dilution according to Srere [29]. All preparation steps were performed at 4 °C. Enzyme activities were assayed with an Olli-C analyser (Kone Oy, Espoo, Finland).

### 2.8. Determination of Myosin Heavy Chain Profile

The myosin heavy chain (MHC) isoform composition (MHC I, MHC IIa, MHC IIx) in muscle homogenate was determined by SDS-PAGE gel electrophoresis using a Bio-Rad Protean II Xi vertical slab gel system (Bio-Rad, Hercules, CA, USA) [25].

### 2.9. Adipokine Assays from Serum Samples

Adiponectin was measured using sandwich ELISA assay according to the manufacturer’s instructions (B-Bridge International, Inc Mountain View; Santa Clara, CA, USA). IL-6, leptin, MCP-1 and TNF-α were simultaneously measured with a BioRad Bio-Plex 200 System using LINCOplex Human Adipokine Panel B according to the manufacturer’s instructions (Millipore, Billerica, MA, USA) [30].

### 2.10. Blood Chemistry

Plasma glucose was enzymatically analyzed with hexokinase (Olympus System Reagent, Hamburg, Germany), and serum insulin was determined with a radioimmunoassay (Pharmacia, Uppsala, Sweden). The hemoglobin A1c (HbA1c) concentration was assayed using latex immunoagglutination inhibition methodology (DCA, 2000 Reagent Kit, Bayer Corporation, Elkhart, IN, USA). Bennett’s index was calculated with the formula 1/(ln fasting insulin (μU/mL) × fasting glucose (mmol/L) [31]; the Homeostasis Model Assessment and insulin resistance (HOMA-IR) were calculated according to the formula: HOMA-IR = fasting serum insulin (μU/mL) × fasting plasma glucose (mmol/L)/22.5 [32]. McAuley’s index was calculated according to the formula Mffm/I = exp [2.63 − 0.28 ln(fasting insulin/μU/mL) − 0.31 × ln(fasting triglyceride/mmol/L)] [31]. Serum total cholesterol, high-density lipoprotein (HDL)-cholesterol and triglycerides were determined using enzymatic assay methods (CHOP-PAP, Monotest, Boehringer Mannheim, Germany). The concentrations of free fatty acids were determined using enzymatic colorimetric method, and gamma-glutamyl transferase (GT) was determined by an ECCLS method (Olympus System reagent, Olympus, Hamburg, Germany).

### 2.11. Exercise Test and Maximal Oxygen Uptake (VO_2_max)

A 2 min incremental cycle ergometer test until volitional exhaustion or fatigue of the lower limbs was employed to measure the maximal oxygen uptake (VO_2_max). The warm-up loading was 30–40 W for women and 40–60 W for men, depending on the age, size and physical fitness of the subjects. Thereafter, the work rate was increased every second minute with equal increments (10–25 W) throughout the test. The increments were individually determined based on the subject’s physical fitness, so that the maximum work rate would be reached in about 12–15 min [33]. Respiratory gas exchange was measured continuously using a breath-by-breath method. VO_2_max was recorded as the highest averaged value over 30 s at the work rate maximum [31].

### 2.12. Assessment of Dietary Intake

Nutrient intakes were assessed using a dietary analysis program developed by the National Public Health Institute [34]. At baseline and before the 24-month visit, the subjects were asked to complete a three-day food record [35,36].

### 2.13. Statistical Analysis

Data are reported as means ± standard error (SE). Student’s paired t test was used to assess differences within groups (baseline and 2-year intervention) with the Kruskall–Wallis test applied for examining differences between the groups. IBM SPSS statistics software version 27.0 (IBM, Armonk, NY, USA) was used for the statistical analyses. All reported *p* values were based on 2-tailed statistical tests, with a significance level set at *p* < 0.05. Correlation analyses were performed between changes in the weight loss and VO_2_max, levels of adiponectin and leptin, citrate synthase activity, Bennett index, McAuley index and HOMA-IR using the Pearson correlation test. The strength of association according to the Pearson correlation coefficient was considered as weak (0.1–0.39), moderate (0.40–0.69), strong (0.7–0.89) or very strong (0.9–1.00). Cohen D was used to determine the effect size for a statistically significant difference within groups and between groups by estimating eta squared based on the H-statistic (eta^2^[H] = (H − k + 1)/(n − k). The effect size is assessed as follows; small (d = 0.2), medium (d = 0.5) and large (d ≥ 0.8) based on Cohen D; eta squared (η^2^) is evaluated as follows; small (0.01–<0.06), moderate effect (0.06–<0.14 and large effect (≥0.14)

## 3. Results

The exercise–diet intervention induced a significant weight reduction during the 2-year follow-up in whole intervention group (n = 22) (−4.4 ± 5.0); the change was much greater in the responders (−7.7 ± 1.0) than in the non-responders (+0.3 ± 0.6; *p* < 0.001). In addition, waist–hip ratio (WHR) decreased in the responders as compared to baseline values (*p* < 0.01) but not in the non-responders (n.s). There was no difference detected in the training activity between the subgroups during the 2-year exercise–diet intervention (Table 3). The proportions of MHC (myosin heavy chain) I, MHC IIa and MHC IIX were similar at baseline (Table 1). The exercise training increased the proportion of MHC I (*p* < 0.05) and slightly decreased (n.s) the proportion of MHC IIx isoforms in the vastus lateralis muscle in the responders, but this change did not occur in the non-responders. After the 2-year exercise–diet intervention, the proportion of MHC I was significantly higher in the responders than in the non-responders (*p* < 0.05) (Table 3).

### 3.1. Aerobic Performance

The aerobic capacity was at the same level in both subgroups at the baseline (Table 4). The 2-year intervention increased both maximal oxygen uptake values VO_2_max and VO_2_peak (*p* < 0.01 and *p* < 0.001, respectively) and oxidative capacity (CS) (*p* < 0.01) of the muscle tissue in the responders but only VO_2_max (*p* < 0.05) values in the non-responders (Table 4). Our results display a large effect size for the changes in VO_2_max, VO_2_peak and CS in the responder group (d = −1.108, 4.368 and −0.879, respectively).

### 3.2. Dietary Energy Intake

At baseline, the dietary energy intake was similar in the non-responders and responders. Furthermore, there were no statistically significant differences in dietary intake between these groups after the 2-year exercise–diet intervention (Table 4). In the responders, dietary fiber intake (g) significantly increased as compared to baseline values (*p* < 0.05) and dietary fat intake as a percentage of the total daily energy (E%) decreased from 35.1% to 30.6% (*p* < 0.05) (Table 4), but not in the non-responders.

### 3.3. Glucose Metabolism

The metabolic indices, i.e., fasting glucose concentration, 2 h glucose, fasting insulin, 2 h insulin, HbA1c and HOMA-IR at baseline were similar in the non-responders and responders. During the 2-year intervention, the fasting glucose concentration, 2 h glucose, fasting insulin, HbA1c and HOMA-IR significantly decreased in the responders, and 2 h insulin tended to decrease (Table 5). Insulin sensitivity (Bennett and McAuley indices) increased in responders (*p* < 0.01) during the intervention. After the 2-year intervention, fasting glucose levels (η^2^ = 0.24) 2 h insulin levels (η^2^ = −0.09), HOMA-IR (η^2^ = 0.013) were significantly lower in the responders than in the non-responders (Table 5), and the Bennett (η^2^ = 0.23) and McAuley (η^2^ = 0.32) indices were significantly higher in the responders than in the non-responders. Effect sizes (η^2^) were large in insulin sensitivity indices and in fasting glucose levels and weaker in 2 h insulin and HOMA-IR in responders. In the non-responders, we detected no significant changes in the levels of fasting glucose, 2 h glucose, fasting insulin, 2 h insulin, HbA1c or HOMA-IR values (Table 5).

### 3.4. Lipid Metabolism

At baseline, the profiles of serum lipid metabolites were essentially the same in the non-responders and responders. After the 2-year intervention, serum triglycerides (*p* < 0.01) and free fatty acid (*p* < 0.01) levels significantly decreased, whereas HDL cholesterol (*p* = 0.051) levels tended to increase in the responders (Table 5). Total serum cholesterol and low-density lipoprotein (LDL)-cholesterol concentrations decreased in the non-responders during the intervention (*p* < 0.05 and *p* < 0.01, respectively). In addition, total serum cholesterol and LDL-cholesterol values were significantly lower in the non-responders (*p* < 0.05) as compared to the responders at the end of the 2-year intervention (Table 5).

### 3.5. Systemic Concentrations of Adipokines

At baseline, the levels of the adipokines in blood (adiponectin, IL-6, leptin, MCP-1 and TNF-α) were similar in the non-responders and responders (Table 6). The exercise–diet intervention increased systemic adiponectin concentrations by 12% while decreasing the leptin concentration by 27% in the responders, whereas in the non-responders the leptin concentration increased by 15%, and the adiponectin concentration decreased by 4% after the 2-year intervention (Table 6). The differential change in leptin levels between groups was statistically significant (*p* < 0.001, η^2^ = 0.44). There were no statistically significant differences in the systemic concentrations of IL-6, MCP-1 and TNF-α either within or between the groups (Table 6).

### 3.6. Cytokine Gene Expression in Skeletal Muscle

We did not detect any difference in the relative change in gene expressions of INF-γ (*p* = 0.053) or TNF-α between the groups (*p* = 0.382) (Figure 2).

### 3.7. Expression of the Genes for Two Insulin Signaling Mediators in Skeletal Muscle

The expressions of IRS-1 (*p* = 0.305) or GLUT-4 (*p* = 0.234) did not differ between the groups after the 2-year exercise–diet intervention (Figure 2).

### 3.8. AMPK Protein Expression in the Skeletal Muscle

At baseline, the protein expressions of the AMPK-α1 and AMPK-α2 were at same level in the studied subgroups. Due to the intervention, the protein expression of AMPK-α1 tended to increase in the responders (*p* < 0.10) but not in the non-responders (Figure 3A). The expression of the AMPK-α2 protein remained unchanged in both subgroups after the 2-year intervention (Figure 3B).

### 3.9. Correlation Analysis between Changes in Weight Loss (kg), Maximal Oxygen Uptake (VO_2_peak) and Laboratory Parameters

In the non-responders, we detected no correlations between any of the studied parameters (Table 7). However, in the responders, there was a weak correlation between changes in CS and VO_2_peak and strong correlations between changes in the value of the VO_2_peak and Bennett and McAuley indices (Table 7). The changes in the amount of weight lost and the leptin concentrations positively correlated in responders (*p* = 0.662) but not with the VO_2_peak (*p* = 0.141).

### 3.10. Diabetes Incidence

Three incident cases of diabetes were diagnosed during five-year follow-up period, 3 in the non-responders group and none in the responders group.

## 4. Discussion

In this study, at the two-year assessment of an intervention involving an increase in physical activity and a modification of diet, there was evidence of enhanced oxygen uptake (10%), increased oxidative capacity of skeletal muscle (37%) and decreased insulin resistance (60%) in the responders. Furthermore, the increased maximal oxygen uptake positively correlated with the improvements in insulin sensitivity. These improvements were associated with a complete prevention of type 2 diabetes as observed in the responders after a 5-year follow-up, while one out of every three of the non-responders (3/9) was diagnosed with type 2 diabetes.

Obesity and physical inactivity are the main non-genetic risk factors for type 2 diabetes [37]. Although increased physical activity and changes in daily dietary habits are the main lifestyle factors recommended for the prevention of type 2 diabetes [38], after weight loss, only 10% of dieters manage to maintain weight loss in the long-term [39]. On the other hand, the DPS (Diabetes Prevention Study) [23] and several other trials have revealed that lifestyle changes can achieve both significant weight losses and improve glucose tolerance in middle-aged obese subjects with impaired glucose tolerance (IGT) [40,41,42] and that these are accompanied with an increase in the concentration of adiponectin and decrease in those of serum IL-6 and TNF-α [43,44,45]. In our study, we detected no changes in the serum levels of IL-6 or TNF-α or MCP-1 in either group, although serum adiponectin concentrations did increase in the responders. In contrast, the gene expressions of INF-γ or TNF-α in skeletal muscle did not change due to the intervention. The expression of pro-inflammatory genes was reported to be elevated in skeletal muscle cells of diabetic patients and interferon-γ-induced up-regulation leading to a decrease in insulin-stimulated glucose uptake [46].

In the present study, in the individuals classified as responders to the exercise–diet intervention, serum leptin levels were reduced (27%), serum adiponectin concentrations increased (12%) and glucose control improved without any changes in the lipid metabolism. In a review, Klimcakova et al. [47] concluded that as little as a 1% weight reduction would be enough to achieve a significant decrease in the plasma leptin concentration. Consistent with other reports [48,49,50], in the present study, the exercise–diet intervention-induced a reduction in serum leptin levels, and this decline was positively correlated with weight loss in the responders. In the other studies, serum leptin decreased by 37% after weight stabilization due to the consumption of a hypocaloric diet for 16 weeks [51], by 21% after a 3-month dynamic strength training program [52] and by 14% after a 6-month program that combined a hypocaloric diet and moderate physical activity [48]. In the study of Auerbach et al., a strong correlation was evident between changes in fat percentage and plasma leptin levels [53].

In the literature, there is still controversy regarding the effects of regular exercise on adiponectin concentrations. An increase in adiponectin concentration was demonstrated in response to weight loss and glitazone therapy, but not after chronic exercise training [54]. According to Madsen et al. [55], a loss of at least 10% body weight is needed to increase systemic levels of total adiponectin. Nonetheless, many investigators reported that adiponectin levels increased in interventions that induced a major weight loss or a significant reduction of visceral fat; in these trials, it seems that weight loss was more efficient than an exercise-based intervention [56]. However, Fatouros et al. [57] reported that, while the leptin concentration was decreased by low-, moderate- and high-intensity resistance training in inactive obese men, in contrast, the adiponectin concentration rose only in the high-intensity training group. These changes seem to be strongly associated with changes in the ratio of muscle and adipose tissue mass. According to Mujumdar et al. [58], a progressive long-term aerobic training (marathon training) increased the adiponectin concentration in middle-aged, overweight, untrained males and females, without affecting the insulin resistance in either group [58]. The exercise intensity in our trial was not as high as in the studies of Fatouros et al. [57] or Mujumdar et al. [58]. Therefore, it is plausible to conclude that the increase in the adiponectin concentration observed in the present study was due to the combined effects of exercise and diet.

Both adiponectin and leptin are hormonal regulators of AMPK signaling and are generally perceived to exert positive effects on insulin sensitivity [15]. The protein expression of AMPK- α1 was increased after one-month’s endurance training in healthy young males and in type 2 diabetes patients after 6 weeks of strength training [59,60]. However, in our study, the protein expressions of the AMPK-α1 and AMPK-α2 in the skeletal muscle remained unchanged in both groups after the 2-year intervention, and there was no correlation between the changes in the amounts of the AMPKs and CS. In work conducted in rodents, when AMPK is chronically activated, a similar adaptive pattern of an increase in mitochondrial oxidative enzymes was observed. Nevertheless, our study design did not allow to us to investigate either AMPK activity or signaling by this protein in skeletal muscle. However, in skeletal muscle, adiponectin activates both AMPK-α1 and AMPK-α2, while leptin activates only AMPK-α2, resulting in enhanced fatty acid oxidation [61,62,63]. In addition, AMPK also regulates glucose homeostasis. The acute regulation of skeletal muscle glucose transport by exercise is multifactorial and has been attributed to several signaling pathways, including Ca^2+^-calmodulin-dependent protein kinase (CaMK)II, AMPK and nitric oxide [64]. Here, the diet and exercise intervention did not increase the expression of the *GLUT-4 gene* in the skeletal muscle of either the non-responders or the responders. During exercise, AMPK activation occurs as a response to the contraction of fast-twitching muscles, triggering an up-regulation of hexokinase II expression, which subsequently increases *GLUT-4 gene* expression and GLUT-4 translocation [65,66,67]. In addition, when considering the response to exercise, AMPK and calcium/calmodulin-dependent protein kinases (CaMK)II contribute to the regulation of GLUT-4 expression via histone hyperacetylation of the GLUT4 promoter and increase GLUT4 transcriptional activity [68]. In summary, the multifaceted regulation of GLUT-4 may explain the failure to detect any difference in GLUT-4 gene expression between responders and non-responders.

It has been shown that physical exercise increases the GLUT-4 content in both healthy controls and subjects with impaired insulin-stimulated glucose uptake [69,70,71,72]. In addition, in subjects with type 2 diabetes, GLUT-4 is normally expressed in skeletal muscle [73,74]. Moreover, an inhibition of protein synthesis did not counteract the exercise-mediated increase in muscle insulin sensitivity [75]. Therefore, increased GLUT4 translocation, rather than its expression, seems to lie behind the improvements in insulin’s action post-exercise [76]. Consistent with this hypothesis, we did not find any increase in the expression of the GLUT4 gene in the responders, despite their improved glycemic control and insulin sensitivity. While physical exercise may be able increase mRNA of GLUT4 in healthy and type 2 patients, it seems that the increase in GLUT mRNA is transient, returning to a pre-exercise level within 24 h [77]. Our results can be interpreted as follows—our exercise–diet intervention had not increased the gene expression of GLUT4 by the end of the 2-year intervention, although it did seem that the exercise intensity could have been high enough to transiently induce the expression of GLUT4.

The failure of non-responders to achieve a weight loss and their lower training intensity may be responsible for their unchanged glucose control or serum adipokine concentrations. Interestingly, total cholesterol and LDL-cholesterol levels only decreased in the non-responders, despite the fact that marginal improvements in the dietary intake of fat and fiber occurred only in responders. On the other hand, serum levels of triglycerides and free fatty acid values were decreased in the responders but not in the non-responders. The chronic adaptation to exercise reduced the synthesis of triglycerides, which may have been a response to the enhanced lipid oxidation in trained skeletal muscle. Aerobic exercise was reported to reduce triglycerides, but this effect was only evident with an accompanying energy deficit [78]. In addition, the exercise–diet intervention may have evoked reductions in free fatty acid levels, and this can improve the insulin sensitivity in both liver and skeletal muscle. In these tissues, the intracellular accumulation of lipids (diacylglycerol) impaired insulin signaling [79]. As a limitation, our study design, did not allow us to reliably evaluate the individual impact of the exercise or changes in the effect of dietary habits on weight loss or improved glucose control. When considering glucose metabolism in skeletal muscle, the greatest changes due to dietary counselling were assumed to have occurred by 6 months. While this gives a good view of the changes in the regulation of the skeletal muscle metabolism induced by exercise, it does not allow us to separately distinguish the individual effects of either exercise or dietary counselling. In addition, in both groups, rather few participants were women, which needs to be considered while interpreting the results. Although we did not observe any difference between female and male subjects in the changes occurring in any of the parameters, because of the small sample size, we could not perform a more detailed statistical analysis. Dietary intake and the total amount of exercise did not significantly differ between responders and non-responders, but the intensity of the exercise might have been higher in responders (based on their increased citrate synthase activity), and this could contribute, at least partly, to these favorable effects, together with dietary modifications such as increased fiber intake and decreased fat intake. There was no difference in the total duration of physical activity, but the amount of exercise training in responders that consisted of two hours aerobic exercise and one hour strength and power exercise per week may be an optimal strategy to improve metabolism and achieve greater caloric expenditure. It seems that the sustained weight loss achieved by dietary modifications with a combination of aerobic and power-type strength exercises is likely to be the major contributor to the improved glucose control evident in the responders. There can be at least three possible explanations to account for the improved glucose control in the responders. The exercise–diet intervention: (1) reduced body weight with decreased adipose tissue mass resulting in a change to a more favorable adipokine profile; (2) the increased oxidative capacity of skeletal muscle coupled with increased fatty acid oxidation in the skeletal muscle, which reduces leptin resistance and increases adiponectin concentrations; and (3) the reduced level of the hepatic lipid content and weight-loss-related improvements of HOMA-IR, triggering decreased glucose production in liver and a loss of intra-hepatic lipids.

While this study had a strong focus on adipokines and cytokines in middle-aged obese subjects with impaired glucose tolerance, it is important to consider the clinical significance of the findings since dietary counselling and physical activity were the main approaches applied in the prevention and treatment of type 2 diabetes. This two-year exercise–diet intervention exerted a significant effect on fasting glucose levels and insulin sensitivity in responders. Insulin sensitivity is not a basic or standard clinical measurement, but it is an important health outcome that can be easily calculated using routine laboratory values. The extent of weight loss was also clinically significant in the responders (>5%), and together with the improvements of maximal oxygen uptake (large effect size), these changes can be considered to underpin the positive alterations in glucose metabolism evident in the responders. It has been claimed that even a minor weight loss should be enough to significantly decrease plasma leptin levels [47], and in our study, the observed effect size of the changes of leptin was large. The weight loss is an important factor accounting for the decrease in the leptin levels, i.e., it correlated strongly with changes in the serum leptin concentration in responders leading to decreased leptin resistance. The observed effect size shows that successful weight loss and improved aerobic capacity can achieve clinically significant changes in glucose control.

## 5. Conclusions

In conclusion, an exercise–diet intervention decreased serum leptin levels, increased serum adiponectin concentrations and improved glucose control without affecting cholesterol metabolism or GLUT-4 expression in the skeletal muscle specimens obtained from the individuals who responded well to the intervention.

## Figures and Tables

**Figure 1 ijerph-19-08327-f001:**
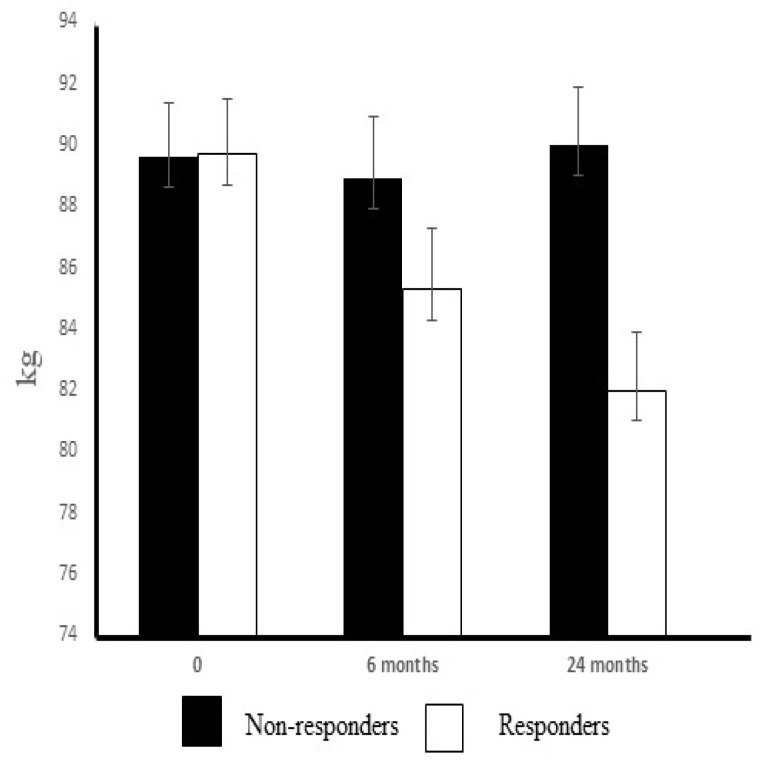
Weight-loss during the 2-year intervention in the subgroups of responders and non-responders in the IGT subjects.

**Figure 2 ijerph-19-08327-f002:**
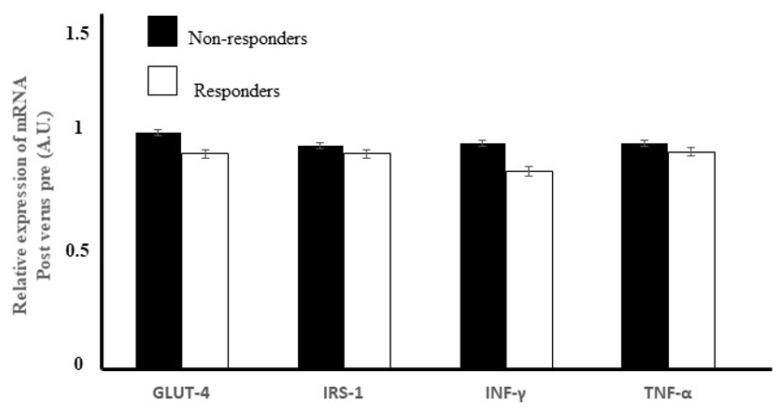
Relative changes in the gene expressions of INF-γ, IRS-1, GLUT-4 and TNF-α after the 2-year exercise–diet intervention in the skeletal muscles of the IGT subjects.

**Figure 3 ijerph-19-08327-f003:**
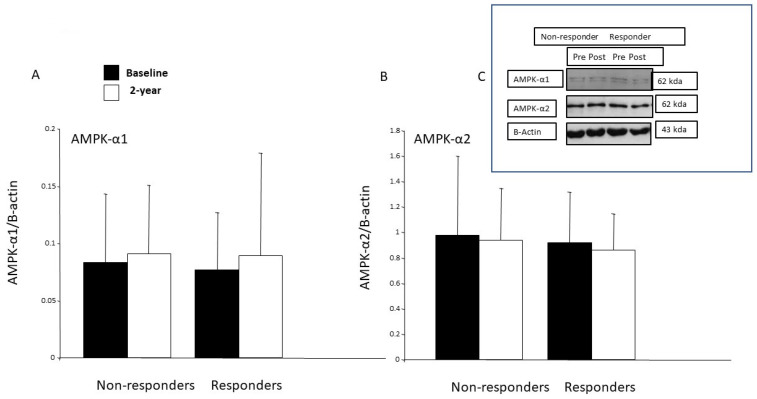
Protein expressions of AMPK- α1 (**A**) and AMPK-α2 (**B**) in skeletal muscle of IGT subjects. (**C**) Representative Western blots, using anti-AMKP-α1 and AMPK-α2 and beta-actin antibodies from skeletal muscle biopsies of non-responders and responders at baseline (0) and after the 2-year intervention.

**Table 1 ijerph-19-08327-t001:** Baseline characteristics of the subjects with impaired glucose tolerance (IGT) subdivided into the non-responders and responders.

Characteristics	Non-Responders	Responders *p* Value	
*n* = (Female/Male)	9 (2/6)	13 (4/9)	n.s
Age (years)	57.1 ± 2.3	55.1 ± 2.0	n.s
Height (cm)	174.2 ± 1.5	172.6 ± 2.2	n.s
Weight (kg)	89.7 ± 1.8	89.7 ± 2.9	n.s
BMI	29.5 ± 0.6	30.1 ± 0.7	n.s
WHR	0.96 ± 0.02	0.96 ± 0.02	n.s
*Aerobic performance capacity*			
VO_2_max(L/min)	2.37 ± 0.22	2.26 ± 0.18	n.s
VO_2_peak (mL/kg^−1^/min)	26.7 ± 2.7	26.2 ± 1.9	n.s
*Muscle fiber type composition*			
MHC I	38.0 ± 3.0	37.9 ± 3.3	n.s
MHC IIa	41.6 ± 2.7	43.4 ± 3.2	n.s
MHC IIx	21.0 ± 2.7	18.7 ± 2.8	n.s
*Dietary intake*			
Energy intake, kcal/day	1823 ± 143	1925 ± 169	n.s
Alcohol, E%	2.6 ± 1.3	4.5 ± 1.2	n.s
Carbohydrates, E%	44.7 ± 2.0	42.5 ± 1.8	n.s
Fat, E%	35.6 ± 1.6	35.1 ± 1.1	n.s
Protein, E%	17.1 ± 1.2	17.9 ± 0.7	n.s
Dietary fiber, g/day	22.3 ± 2.1	19.7 ± 1.9	n.s

Data are presented as means ± SE. Kruskal–Wallis test between groups. n.s non-significant, BMI: body mass index, WHR: waist-hip-ratio, MHC: myosin heavy chain, VO_2_max: maximal oxygen uptake, VO_2_peak: peak oxygen uptake, E% = percent of energy intake.

**Table 2 ijerph-19-08327-t002:** List of primers and probes used in the RT-PCR assay.

GAPDH	
Probe	5′-ACCAGGCGCCCAATACGACCAA-3′
Forward primer:	5′-GTTCGACAGTCAGCCGCATC-3′
Reverse primer:	5′-GGAATTTGCCATGGGTGGA-3′
GLUT-4	
Probe:	5′-CTC-AGC-CAG-CAC-TCC-AGA-AAC-ATC-GG-3′
Forward primer:	5′-AAG-AGT-CTG-AAG-CGC-CTG-ACA-3′
Reverse primer:	5′-CAG-CTT-CCG-CTT-CTC-ATC-CT-3′
INF-γ	
Probe:	5′-TGCTGGCGACAGTTCAGCCATCAC-3′
Forward primer:	5′-CTCGAAACAGCATCTGACTCCTT-3′
Reverse primer:	5′-TGTCCAACGCAAAGCAATACA-3′
IRS-1	
Probe:	5′-AAACCCATTCTCTCATGACACGGTGGTG-3′
Forward primer:	5′-TCTCCACCCAACGTGAACAGT-3′
Reverse primer:	5′-CTGCATAAACTTCATCTTCAACCTTAAG-3′
TNF-α	
Probe:	5′-CATCTTCTCGAACCCCGAGTGACAAGC-3′
Forward primer:	5′-TGGCCCAGGCAGTCAGA-3′
Reverse primer:	5′-GGTTTGCTACAACATGGGCTACA-3′

GAPDH: glyceraldehyde-3-phosphate dehydrogenase, INF-γ: interferon-γ, GLUT-4: glucose transporter protein-4, IRS-1: insulin receptor substrate-1, TNF-α: tumor necrosis factor.

**Table 3 ijerph-19-08327-t003:** Exercise frequency and changes in muscle fiber composition occurring during 2-year intervention in the IGT subjects.

	Non-Responders	Responders
Exercise frequency (hours/week)		
Moderate and heavy aerobic training	1.2 ± 0.3	1.3 ± 0.3
Strength and power training	2.4 ± 0.6	1.3 ± 0.2
Walking and lighter aerobic training	1.7 ± 0.6	2.3 ± 0.4
Total amount of training	5.4 ± 0.8	4.9 ± 0.8
Muscle fiber composition%		
ΔMHC I	−1.5 ± 1.4 ^#^	3.8 ± 1.7 *
ΔMHC IIa	1.3 ± 2.0	1.0 ± 2.6
ΔMHC IIx	0.2 ± 2.0	−4.9 ± 2.8

Data are presented as means ± SE; * *p* < 0.05 within groups using paired Student *t*-test; ^#^
*p*< 0.05 between groups using Kruskal–Wallis test.

**Table 4 ijerph-19-08327-t004:** Aerobic performance capacity and dietary intake at baseline and after the 2-year in IGT subjects.

	Non-Responders		Responders	
	Baseline	2-Year	Baseline	2-Year
Aerobic performance capacity				
CS (μmol/g wet weight muscle tissue)	22.9 ± 3.4	25.6 ± 3.2	22.6 ± 2.0	31 ± 3.2 **
VO_2_max (L/min)	2.37 ± 0.22	2.52 ± 0.22 *	2.26 ± 0.18	2.37 ± 0.17 **
VO_2_peak (mL//kg/min)	26.7 ± 2.7	28.1 ± 2.5	26.2 ± 1.9	28.8 ± 1.9 ***
Dietary intake				
Energy intake, kcal/day	1823 ± 143	1916 ± 169	1925 ± 169	1886 ± 136
Alcohol, E%	2.6 ± 1.3	2.1 ± 1.0	4.5 ± 1.2	5.4 ± 2.1
Carbohydrates, E%	44.7 ± 2.0	47.0 ± 2.6	42.5 ± 1.8	46.1 ± 2.1
Fat, E%	35.6 ± 1.6	33.2 ± 1.3	35.1 ± 1.1	30.6 ± 1.1 *
Protein, E%	17.1 ± 1.2	17.6 ± 1.3	17.9 ± 0.7	17.9 ± 0.7
Dietary fiber, g/day	22.3 ± 2.1	23.2 ± 2.8	19.7 ± 1.9	22.7 ± 1.4 *

Data are presented as means ± SE; * *p* < 0.05, ** *p* < 0.01, *** *p* < 0.001 within groups using paired Student *t*-test. CS: citrate synthase, VO_2_max: maximal oxygen uptake, E% = percent of energy intake.

**Table 5 ijerph-19-08327-t005:** Effects of a 2-year exercise–diet intervention on parameters reflecting glucose lipid metabolism in the IGT subjects.

	Non-Responders			Responders		
	Baseline	2-Year	Changes	Baseline	2-Year	Changes
Fs-Ins (μU/mL)	16.7 ± 2.7	17.0 ± 3.0	0.3 ± 2.7	16.4 ± 2.1	11.2 ± 1.3 **	−5.2 ± 1.3
2 h Ins (μU/mL)	96 ± 23	81 ± 22	−15 ± 19 ^#^	97 ± 33	45 ± 90	−55 ± 30
HbA1c (%)	5.6 ± 0.1	5.4 ± 0.1	−0.2 ± 0.1	5.8 ± 0.1	5.3 ± 0.1 ***	−0.5 ± 0.1
Fs-Gluc (mmol/L)	6.0 ± 0.1	6.1 ± 0.2	0.1 ± 0.1 ^##^	6.1 ± 0.1	5.6 ± 0.1 ***	−0.5 ± 0.1
2 h Gluc (mmol/L)	8.1 ± 0.3	7.0 ± 0.8	−1.1 ± 0.9	7.4 ± 0.4	6.0 ± 0.4 *	−1.4 ± 0.5
Bennett I	0.51 ± 0.02	0.52 ± 0.02	0.01 ± 0.01 ^##^	0.51 ± 0.02	0.57 ± 0.02 **	0.06 ± 0.01
HOMA-IR	4.5 ± 0.8	4.6 ± 0.9	0.1 ± 0.8 ^#^	4.5 ± 0.6	2.8 ± 0.3 **	−1.7 ± 0.4
McAuley I	5.8 ± 0.3	5.7 ± 0.4	−0.2 ± 0.2 ^##^	5.6 ± 0.4	6.5 ± 0.3 **	1.0 ± 0.2
Cholesterol (mmol/L)	5.3 ± 0.3	4.9 ± 0.2 *	−0.4 ± 0.2 ^#^	5.6 ± 0.2	5.5 ± 0.3	−0.1 ± 0.1
LDL (mmol/L)	3.5 ± 0.3	3.0 ± 0.2 **	−0.5 ± 0.12 ^#^	3.7 ± 0.3	3.5 ± 0.3	−0.1 ± 0.12
HDL (mmol/L)	1.10 ± 0.10	1.14 ± 0.10	0.05 ± 0.03	1.14 ± 0.09	1.26 ± 0.09	0.11 ± 0.05
Triglyceride (mmol/L)	1.54 ± 0.15	1.76 ± 0.21	0.22 ± 0.06 *^,##^	1.96 ± 0.24	1.55 ± 0.18 *	−0.42 ± 0.13
FFA (mmol/L)	0.68 ± 0.07	0.47 ± 0.07	−0.21 ± 0.11	0.69 ± 0.07	0.47 ± 0.04 **	−0.22 ± 0.06
γ-GT (IU/L)	36.3 ± 3.2	31.4 ± 5.0	−4.9 ± 3.1	40.9 ± 5.1	31.5 ± 4.5 **	−9.4 ± 2.9

Data are presented as means ± SE; * *p* < 0.05, ** *p* < 0.01, *** *p* < 0.001 within groups using paired Student *t*-test; ^#^
*p* < 0.05, ^##^
*p* < 0.01 between groups using Kruskal–Wallis test. Fs-Ins: fasting insulin, HbA1c: Haemoglobin A1c, I: index, HOMA-IR: Homeostasis Model Assessment, insulin resistance, Fs-Gluc: fasting glucose concentration, 2 h Gluc: glucose concentration at 2 h of oral glucose tolerance test, LDL: low-density lipoprotein, HDL: high-density lipoprotein, FFA: free fatty acids, γ-GT: γ-glutamyltransferase.

**Table 6 ijerph-19-08327-t006:** Effects of a 2-year exercise–diet intervention on the parameters depicting adipokines and cytokines in IGT subjects.

	Non-Responders			Responders		
	Baseline	2-Year	Changes	Baseline	2-Year	Changes
Adiponectin (μg/mL)	6.8 ± 1.5	−0.3 ± 0.5	7.6 ± 1.3	8.5 ± 1.2 *	1.0 ± 0.5	7.1 ± 1.2
IL-6 (pg/mL)	3.5 ± 1.9	3.7 ± 2.1	0.2 ± 0.3	2.9 ± 2.0	2.2 ± 1.4	−0.7 ± 0.6
Leptin (pg/mL)	11.7 ± 3.1	13.4 ± 3.8	1.7 ± 1.0	14.2 ± 3.6	10.4 ± 3.2 **	−3.8 ± 1.0 ^##^
MCP-1 (pg/mL)	348 ± 52	307 ± 43	−41 ± 24	279 ± 35	291 ± 43	11 ± 29
TNF-α (pg/mL)	5.4 ± 0.9	5.8 ± 1.2	0.4 ± 0.5	4.5 ± 0.7	4.6 ± 0.6	0.03 ± 0.5

Data are presented as means ± SE; * *p* < 0.05, ** *p* < 0.01 within groups using paired Student *t* -test; ^##^
*p* < 0.01 between groups using Kruskal–Wallis test. IL-6: interleukin-6, MCP-1: monocyte chemoattractant protein-1, TNF-α: tumor necrosis factor.

**Table 7 ijerph-19-08327-t007:** Correlation coefficients between weight loss and maximal oxygen uptake (VO_2_peak) and laboratory parameters.

	Non-Responders		Responders	
	Weight Loss	VO_2_peak	Weight Loss	VO_2_peak
Change: adiponectin	−0.477	0.258	−0.514	−0.178
Change: citrate synthase	0.152	−0.330	0.079	0.040 *
Change: leptin	0.171	−0.347	0.662 *	0.141
Change: Bennett	−0.129	0.540	−0.506	0.831 **
Change: HOMA-IR	0.327	−0.632	0.220	−0.520
Change: McAuley	0.063	0.457	−0.344	0.890 **

Pearson correlation test, * *p* < 0.05, ** *p* < 0.01.

## Data Availability

Not applicable.

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
