# Peer review of "Improved Aerobic Capacity and Adipokine Profile Together with Weight Loss Improve Glycemic Control without Changes in Skeletal Muscle GLUT-4 Gene Expression in Middle-Aged Subjects with Impaired Glucose Tolerance"

_ijerph, 2022, doi:10.3390/ijerph19148327_

Round 1
Reviewer 1 Report
Many reported results were not described in the methods section. Tables were badly format. Authors failed to provide a clear rationale for the study. Results in the tables should be shown also as delta, as groups started the intervention with numerical differences in the variables assessed. The reader can be confused with the use of the word “expression” (gene or protein?). mRNA expression is usually used for acute exercise studies, not long-term ones, especially when one considers signaling pathways. The reporting of results is very confusing and a thorough English review is urgently needed, especially in the discussion. The discussion is long and authors fail to discuss their results in the context of the existing literature. Discussion is unacceptable as it is.
- In the title, abstract and throughout the text, authors should clearly state that gene expression was assessed for GLUT-4 and interferon gamma. The reader can be led to believe authors assessed protein expression when they read in the title and abstract the term “expression” only.
- The title mentions nothing about adipokines, although this is the first aim in the abstract.
- Sentence in lines 19-21 needs rewriting.
- Define ING and GLUT4 acronyms. Define all acronyms at first entrance.
- In the main text interferon gamma acronym is different from the abstract.
- Line 24: improving or improved?
- Authors assess gene expression, but do not introduce this in the Introduction.
- Authors do not talk about interferon gamma in the Introduction.
- Line 35: no comma after “revealed that”
- Line 36-39: is it only the adipose tissue that releases IL-6 and TNF-alpha? Is it correct to refer to them as “adipokines”?
- Line 41: AMPK should be in parentheses and no need for “(AMP-activated protein kinase)”.
- Line 47: “and therefore IS known…”
- Line 49: “physical exercise stimulateS”
- Line 61: what do authors mean by “AMPK activity at the whole-body level”?
- Lines 71-77: I disagree that the authors explored the mechanisms. They rather reported responses to a treatment, not the mechanisms involved.
- Figure 1: report STD or SEM
- Table 1: format this table. Columns are not aligned.
- Table 1: In VO2max, “2” should be subscript
- Table 1: it is either L/min or L.min-1, not L/min-1.
- Table 1: what does “ind” mean after VO2max?
- Table 1: please provide statistical comparisons for responders vs non-responders.
- Table 1: please provide the definitions for the acronyms in table legend.
- Table 3: please align columns.
- Table 3: please double check the p value for the comparison between strength and power training. It seems too large not be significant. The same goes for walking and lighter training.
- Nothing is mentioned in the methods about MHC assessment.
- Table 3: the text mentions MHC IIx, but the table only shows IIaIIx.
- Line 222: if MHC IIx was “slight decreased” the values in the table should be negative.
- Table 4: align columns
- Table 4: what is “aerobic performance capacity”?
- Table 4: In VO2max, “2” should be subscript
- Table 4: what does “ind” mean after VO2max?
- Table 4: Citrate synthase activity should be normalized by protein concentration of the homogenate used, not wet weight.
- Table 5: was an OGTT performed? Nothing is mentioned in the methods.
- Table 5: Define acronyms in the table legend
- Lines 273-6: if changes were not statistically significant, do not mention they increased or decreased.
- Figure 2: I am confused with the unit (arbitrary units) used for mRNA. It should be fold change. Please, cite the method for calculating mRNA expression.
- Figure 2: report STD or SEM.
- Figure 2: use dot, not comma
- Figure 2: no legend for statistical analysis.
- Line 290: authors report the decrease in interferon was not significant, but then in line 292 report there was a decrease. This is very confusing. Actually, this whole paragraph needs a thorough revision.
- Figure 3: please provide STD; review the unit used; provide legend for statistical analysis.
- Figure 3: what is the idea of measuring mRNA expression of GLUT-4 or IRS-1 in this study?
- Figure 4: provide blots
- Figure 4: review unit: report as AMPk/beta actin
- Figure 4: provide legend for statistical analysis
- I see no rationale for measuring AMPk protein content in a chronic study. Please provide the rationale.
- Lines 327-329: review sentence, as it is meaningless and needs grammar review
- Lines 331-333: review sentence, as it is meaningless
- Line 335: DPS?
- Line 352: did authors run a correlation analysis?
- Line 354: parentheses after 45
- Line 381-90: authors discuss AMPk in the context of acute exercise-induced glucose uptake. Please, discuss within the context of chronic studies.
- Line 392-3: please provide a reference that showed that AMPk activates mRNA gene expression, as ref 59 does not do that.
- The discussion of GLUT-4 mRNA expression is not done within the context of chronic studies.
- Line 400: was there a statistical difference for strength training time between groups?
- Line 417: insulin resistance augments lipolysis. So how can improved insulin sensitivity decrease triglyceride synthesis?
Reviewer 2 Report
The study comes up with an interesting hypothesis. The authors evaluated the role of adipokines in regulation of glucose metabolism in middle-aged obese subjects having impaired glucose tolerance during a long-term exercise and dietary intervention. The manuscript focuses an interesting topic that is worth to be published, minor revision is required.
-English needs to be improved
-In the Materials and Methods line 91 could the authors define the sex of the subjects enrolled in the study?
In the discussion could the authors define the possible differences between men and women in the response at glucose tolerance? it would be very interesting to highlight this, as women have a different response to exercise.
Reviewer 3 Report
IJERPH 1685704
Increased aerobic capacity and weight loss improve glycemic control without skeletal muscle Glut-4 expression changes in middle-aged subjects with impaired glucose tolerance.
______________________________________
Lines 79-80: This RCT must report the website and number of registered for the trial. Please, include it here.
Lines 91-92: To mention how the randomization process was done. Include here.
Figure 1: To show data about central tendency plus dispersion.
Table 1: To report how was measured the dietary intake variable.
Line 108: Intervention: The protocol should be more explained (time, intensity, duration, volume, % of maximal resistance of force production, etc.). What muscle groups were trained? How many days the participants trained ?. This item is crucial to understand whether the changes obtained at primary outcomes were by exercise-induced changes at muscle level or not.
Line: Real-time "PCR"…and previously (line 43), this term was reported as RCP.
Figures 2A and 2B: Please, modify the parenthesis used to inform statistical difference (see international guidelines to show results by images).
Figure 2B: To add the variable's name at the "y" axis.
Figures: Please, include central tendency and dispersion variables in the images used to show the data. The quality of the images must be improved.
It is mandatory in the RCT to inform the statistical difference and effect size of the changes obtained. Please, include these data (and include them in the discussion item).
Round 2
Reviewer 3 Report
The effect size interpretation did not include in the statistical analyses and discussion section in this new version. This is mandatory in this type of study because helps researchers find the minimal important difference (MID) in the main outcome associated with training intervention.
Author Response
The effect size interpretation did not include in the statistical analyses and discussion section in this new version. This is mandatory in this type of study because helps researchers find the minimal important difference (MID) in the main outcome associated with training intervention.
We thank the reviewer for the favorable comment. In the new revision of our manuscript, we included a detailed effect size interpretation. In addition, our manuscript was extensively revised for the English language grammar and spelling by a language editor who is a native English speaker and a researcher.